# An Efficient Method to Compensate Receiver Clock Jumps in Real-Time Precise Point Positioning

**Shaoguang Xu** [1,*], **Jialu Long** [2], **Jinling Wang** [3] and **Wenhao Zhang** [3]

1 Faculty of Geoscience and Environmental Engineering, Southwest Jiaotong University, Chengdu 611756, China
2 School of Natural Resources and Surveying, Nanning Normal University, Nanning 530001, China
3 School of Civil and Environmental Engineering, The University of New South Wales, Sydney 2052, Australia
* Correspondence: shaoguangxu@swjtu.edu.cn; Tel.: +86-158-8225-6854

**Abstract:** In global navigation satellite systems (GNSSs)-based positioning, user receiver clock jump is a common phenomenon on the low-cost receiver clocks and can break the continuity of observation time tag, carrier phase and pseudo range. The discontinuity may affect precise point positioning-related parameter estimation, including receiver clock error, position, troposphere and ionosphere parameters. It is important to note that these parameters can be used for timing, positioning, atmospheric inversion and so on. In response to this problem, the receiver clock jumps are divided into two types. The first one can be expressed by the carrier phase and pseudo range having the same scale jump, and the second one is that they are having different scale jumps. For the first type, if a small priori variance of receiver clock error is provided, it can affect the accuracy of ionospheric delay estimation both in static and kinematic mode, while in the latter mode, it also affects position estimation. However, if large process noise is provided, numerical problems may arise since other parameters' process noises are usually small, it is proposed to use the single point positioning with pseudo ranges to provide a priori value of receiver clock error, and an empiric value is assigned to its prior variance, this handle can avoid the above problems. For the second type, instead of compensating so many raw observations in the traditional methods, it is proposed to compensate the ambiguities at the clock jump epochs only in a new method. The new method corrects the Melbourne–Wubbena (MW) combination firstly in order to avoid the misjudging of cycle slips for current epoch, and the second step is to compensate the corresponding ambiguities, then, after Kalman filtering, the MW and its mean should be corrected back in order to avoid the misjudging of cycle slips at the next epoch. This approach has the advantage of handling the clock jump epoch-wise and can avoid correcting the rest of the observations as the traditional methods used to. With the numerical validation examples both in static and kinematic modes, it shows the new method is simple but efficient for real time precise point positioning (PPP).

**Keywords:** receiver clock jump; precise point positioning; receiver clock error; ambiguity; Melbourne–Wubbena combination





## 1. Introduction

The clock is important for global navigation satellite systems (GNSSs) positioning, since the GNSS observation is measured with satellite and receiver clocks' difference multiplied by the light speed [1,2]. In order to archive the highest positioning accuracy, the satellites in space are installed with atomic clocks which are very stable. At the user end, most of the receivers on the ground are usually equipped with low-cost clocks except some special sites for time datum or other special use. Currently, in order to keep consistent with global positioning system (GPS) time, there are satellite clock error and receiver clock error for satellite and receiver during the positioning [1,2]. The clock's character is shown on either satellite clock error or receiver clock error. For double difference positioning, the clock

error for satellites and receivers can be eliminated [3]. For another positioning—precise point positioning (PPP)—the satellite clock errors are as known together with the satellite coordinates from the network-based products such as those from the International GNSS Service (IGS), but the receiver clock error needs to be estimated together with receiver coordinates [4,5].

In the PPP, it consumes time to obtain mm or cm accuracy in static and cm or dm accuracy in kinematic mode. There are some factors such as the number of satellites, frequent cycle slips, and observation interval that may affect the convergence time [6]. There is another factor, named receiver clock jump, which also affects the PPP's results. Receiver clock jump is a common phenomenon on the low-cost receiver clock, and it usually breaks the continuity of pseudo range or carrier phase observation with 1 ms (millisecond), sometimes it even may destroy the continuity of the observation time tag, and in the worst situation it leads to the inconsistency between pseudo range and carrier phase [7]. Clock jumps should be carefully treated, and in some cases they are mistaken as cycle slips, which lead to initializing the new ambiguity parameters and increase the convergence time. Despite some of the current GNSS observations downloaded from IGS data centers or other sources having been corrected before uploading to keep the pseudo range and carrier phase consistent, those who want to process the previous observations or real-time stream still have to deal with this problem; also, the old receivers which have clock jump phenomenon such as the Trimble 5700 may be still in operation [8,9]. There is a famous program named "*ClockPrep*" which was developed by *Freymueller* and can be used for post processing with all kinds of clock jumps, but this program only matches with the GIPSY software (IGSMAIL-4318). There are also some other research works for receiver clock jumps, but most of them focus on the clock jump detection, its effect on positioning, and observation repairing [7,10,11]. It should be noted that the observation repairing work involves the epochs from the time of clock jump's first occurrence to the end of observation. Additionally, due to the uncertainty of the receiver clock error's variation, it usually modelled as white noise or random walk [12]. However, if unreasonable process noise is provided for receiver clock, it degrades the results of not only position but also troposphere and ionosphere estimation when the receiver clock jumps happen. This study provides a simple but efficient and robust method to deal with the receiver clock jumps of corrected observations and inconsistent observations; for the latter ones, there is no need to compensate so many original observations, while the new method only needs additional handling at the clock jump epochs. The new method divides the clock jumps into two types, influences of each type of clock jump on position, tropospheric delay and ionospheric delay are analyzed in static and kinematic PPP, and at the same time, the feasibility and effectiveness of the new method are validated.

In the rest this paper, the detail of the new method is first presented, then the next Section shows the numerical validation for the new method based on examples, and the final Sections present the discussion and concluding remarks.

## 2. Method

### 2.1. Theory

For real time GNSS PPP, sequential least squares and Kalman filter estimations are the two most used methods; they are popular for the advantage of less computation burden since only information of adjacent epochs is required, and these two methods are essentially equivalent. This article takes the Kalman filter as an estimator, and it can be divided into two steps: the first one is called the Kalman filter prediction, as seen in Equation (1), and the second one is named the Kalman filter update, as seen in Equation (2).

$$\begin{aligned} X_{k+1}^- &= \Phi_{k+1}\widetilde{X}_k \\ P_{k+1}^- &= \Phi_{k+1}\widetilde{P}_k\Phi_{k+1}^T + Q_{k+1} \end{aligned} \tag{1}$$

$$G_{k+1} = P_{k+1}^- H^T (H P_{k+1}^- H^T + R_{k+1})^{-1}$$
$$\widetilde{X}_{k+1} = X_{k+1}^- + G_{k+1}(Z_{k+1} - H X_{k+1}^-) \tag{2}$$
$$\widetilde{P}_{k+1} = (I - G_{k+1}H)P_{k+1}^-$$

where $X_{k+1}^-$ denotes the predicted value of parameters to be estimated at epoch $k + 1$, and the $\widetilde{X}_k$ denotes the updated value of estimated parameters at epoch $k$; $\Phi_{k+1}$ is the state transition matrix of all parameters from epoch $k$ to $k + 1$; $P_{k+1}^-$ and $\widetilde{P}_k$ are the corresponding prior and post covariance matrix of all parameters at the epoch $k + 1$ and $k$ respectively; the super script "$T$" means transposition; $Q_{k+1}$ is the process noise matrix of all parameters; $G_{k+1}$ is the gain matrix; $H$ is the design matrix at epoch $k + 1$; $R_{k+1}$ denotes the measurement noise matrix at current epoch; $\widetilde{X}_{k+1}$ is the updated value of all parameters at current epoch; $Z_{k+1}$ means the matrix of observation minus computation; $\widetilde{P}_{k+1}$ is the updated covariance matrix of all parameters; $I$ is a identify matrix whose dimension depends on parameter number. The parameters $X$ to be estimated in undifferentiated and uncombined PPP for dual frequency can be expressed as follows:

$$X = [x \quad y \quad z \quad t_r \quad ZTD \quad SION^1 \quad \cdots \quad SION^s \quad N_i^1 \quad \cdots \quad N_i^s \quad N_j^1 \quad \cdots \quad N_j^s]^T \tag{3}$$

where $x$, $y$, and $z$ are position parameters which keep constant in static mode and vary every epoch in kinematic mode; $t_r$ denotes receiver clock error which may suffered from clock jump; $ZTD$ is a shorten for zenith total delay; $SION^s$ is the slant ionospheric delay for satellite $s$, $N_i^s$, and $N_j^s$ denote the ambiguities of satellite $s$ on frequency $i$ and $j$, respectively.

Before Kalman filtering, the observation should be preprocessed to obtain clean data, and this preprocessing work includes the clock jump detection, especially when the carrier phase and pseudo range are inconsistent, and one of them should be corrected to keep consistent with other. During the Kalman filter, reasonable noise should be given for each parameter in order to obtain a better result, and the essence of process noise setting is to restrict variation of parameters; among all the parameters, $t_r$ usually shows the most uncertainty, while for other parameters, the process noise of ambiguity is 0, and the left ones do not vary dramatically.

As noted in the Introduction Section, the receiver clock jump may exist in different forms. In the PPP, the quantities $q_{clkjp}$ directly related to the clock jump are as follows:

$$q_{clkjp} = \begin{bmatrix} t_{obs} & P_i & L_i & t_r & N_i & MW \end{bmatrix} \tag{4}$$

where $t_{obs}$ is the observation record time tag, $P_i$ and $L_i$ denote the pseudo range and carrier phase in distance of frequency $i$ without clock jump, $N_i$ denotes the corresponding distance of ambiguity in $L_i$ which keeps constant if no cycle slip or clock jump occurring, $MW$ is the famous Melbourne–Wubbena combination, and it can be expressed as follows [13,14]:

$$
\begin{aligned}
MW &= \frac{f_i(L_i+\Delta L_i) - f_j(L_j+\Delta L_j)}{f_i - f_j} - \frac{f_i(P_i+\Delta P_i) + f_j(P_j+\Delta P_j)}{f_i + f_j} \\
&= \frac{f_i L_i - f_j L_j}{f_i - f_j} - \frac{f_i P_i + f_j P_j}{f_i + f_j} + \frac{f_i \Delta L_i - f_j \Delta L_j}{f_i - f_j} - \frac{f_i \Delta P_i + f_j \Delta P_j}{f_i + f_j}
\end{aligned}
\tag{5}
$$

where $f$ denotes frequency, the $\Delta L_i$ and $\Delta P_i$ are the clock jump values of carrier phase and pseudo range compared to the previous epoch (if no clock jump happening, both of them are equal to zero). For simplicity, other error sources are not further discussed here. If $\Delta L_i$ equates to $\Delta L_j$ and $\Delta P_i$ equates to $\Delta P_j$ Equation (5) can be simplified as follows:

$$MW = \frac{f_i L_i - f_j L_j}{f_i - f_j} - \frac{f_i P_i + f_j P_j}{f_i + f_j} + \Delta L_i - \Delta P_i \tag{6}$$

Indeed, this is a key character of clock jumps on either pseudo range or carrier phase as it has the same scale jump on different frequencies, but it should be noted that pseudo range

and carrier phase may have different scale jumps at the same time. Based on the character, the geometry free (GF) combination in Equation (7) is not affected by clock jump, so it not only can be used for cycle slip detection but also can be used for clock jump detection.

$$GF = L + \Delta L_i - (L_j + \Delta L_j) = L_i - L_j \tag{7}$$

Due to the $N_i$ in $L_i$, the datum of PPP comes from the $P_i$, so the current $N_i$ should be satisfied with the following formulas:

$$N_i = L_i + \Delta L_i - (P_i + \Delta P_i) \tag{8}$$

According to the above equations, the relationship among the related quantities is shown in Table 1. Despite other types of the receiver clock jumps possibly existing, they can be converted to one of the types in the table.

**Table 1.** Quantity states according to receiver clock jumps of different types.

| Type \ Quantity | $t_{obs}$ | $P$ | $L$ | $t_r$ | $N_i$ | $MW$ |
|---|---|---|---|---|---|---|
| 1 | Jump | No jump | No jump | Jump | No jump | No jump |
| 2 | No jump | Jump | Jump | Jump | No jump | No Jump |
| 3 | No jump | Jump | No Jump | Jump | Jump | Jump |
| 4 | No jump | No jump | Jump | No jump | Jump | Jump |

In Table 1, the first three quantities can be considered as independent variables, and the last three can be considered as dependent variables. For the first type, the absolute value of $t_r$ increases with the observation time, which does not affect cycle slip detection. For the second type, the pseudo range and carrier phase have the same scale clock jump at the same time, which is common in the corrected observations, in this situation, it does not affect the $MW$ to detect the cycle slip too. Obviously, the first type and the second type can be converted to each other. For the third type and the last type, both of them change the $MW$ to affect the cycle slip detection, and also the ambiguity changes too based on Equations (6) and (8). Here $P$ and $L$ denote the distance of pseudo range and carrier phase at any frequency and may suffer from clock jump to distinguish the former $P_i$ and $L_i$.

Before the introduction of the new method, all the clock jumps are broadly divided into the following two types:

Type I: The pseudo range and carrier phase have the same scale clock jump, and the time tag's jump belongs to this type too.

Type II: The pseudo range and carrier phase have different scale jumps.

There are two typical examples for type I and II clock jumps, as seen in Figures 1 and 2. In Figure 1, the original L1 and P1 observations for satellite G02 of site CUT0, which is an IGS track station equipped with TRIMBLE NETR9 receiver, have the same scale clock jump, and due to the improved signal tracking technique, the L1 and P1 time series are generally overlapped, and although clock jumps exist, this observation keeps the carrier phase and pseudo range consistent, and the L2 and P2 have similar trends. Not only the G02 has this phenomenon, but also other satellites at corresponding epochs. While in Figure 2, the raw L1 (* denotes that the L1 has a shit of $3.0 \times 10^7$ m) and P1 observations for satellite G24 of site PIXI, which is a local station equipped with TRIMBLE 5700 receiver, have the different scale clock jumps, the carrier phase is smooth and the pseudo range time series have a shape like a sawtooth; for this clock jump of type I, it should be repaired according to Equation (6) in order to avoid the jump of $MW$ and misjudging of the cycle slip detection.

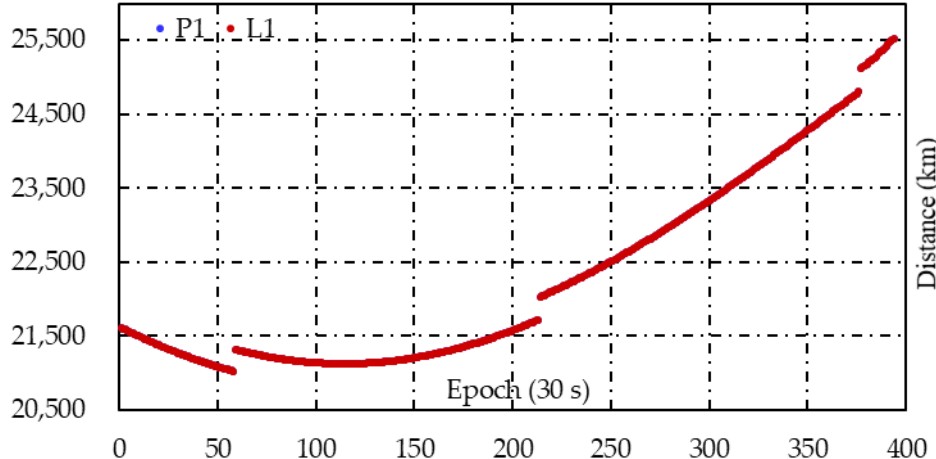

**Figure 1.** Original P1 and L1 observation time series of G02 on DOY 001, 2020, for CUT0 under clock jump of type I.

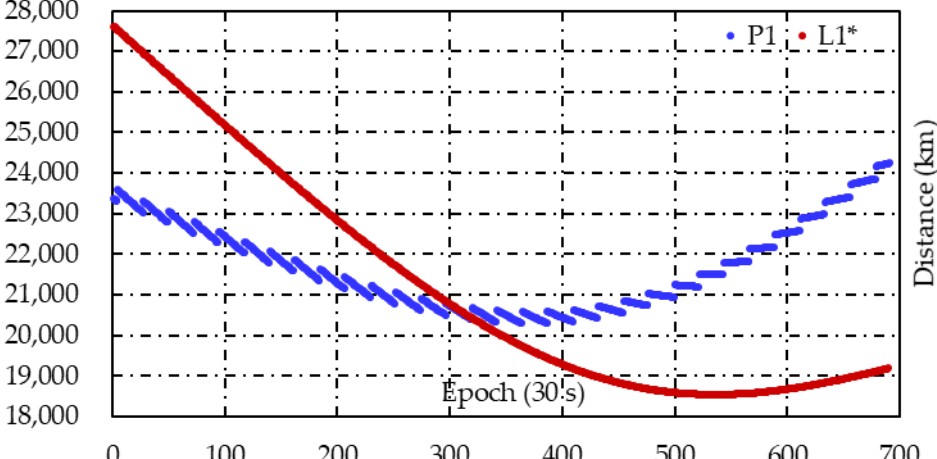

**Figure 2.** Original P1 and L1 observation time series of G24 on DOY 133, 2008, for PIXI under clock jump of type II.

Many IGS station observations downloaded from the IGS data center have the clock jump of type I. Currently, for the clock jump of type II the traditional methods used to correct the original observations [7]. If the clock jump of type II in Figure 2 needs to be converted to the clock jump of type I in Figure 1, the L1 should be corrected with the value (bule curve) in Figure 3 so that then the pseudo range and carrier phase have the same trend. According to Figure 3, once the clock jump happens, all the observations of each frequency for each satellite after this epoch need to be repaired, and the correction value mainly depends on the times of the clock jumps and scale of each jump since the correction value is in the form of steps. It should be noted that someone may want to correct the P1 in Figure 2, but this leads to the unlimited increasing of $t_r$.

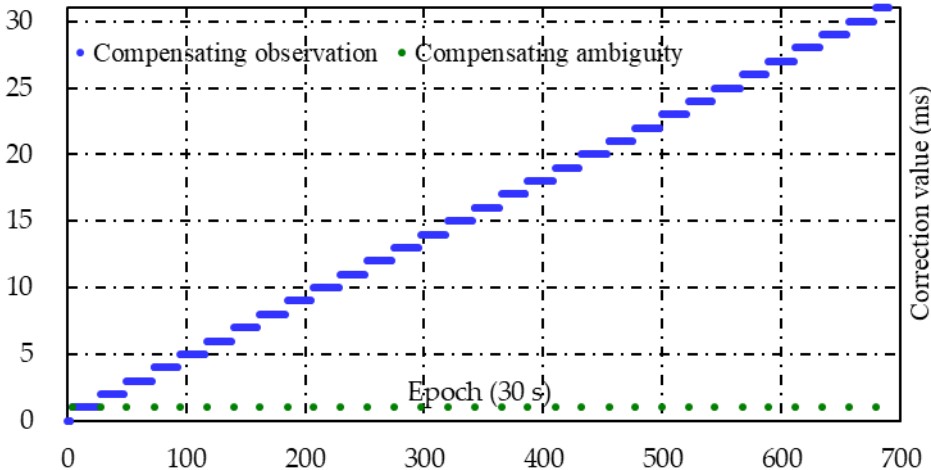

**Figure 3.** Corresponding ms correction to L1 of G24 on DOY 133, 2008, with respect to Figure 2 with traditional and new methods.

Since original observations have direct relations with ambiguity, instead of dealing with so many observations, this type of clock jump may be compensated by corrected ambiguity at the clock jump epochs only. For this situation, Equation (8) can be re-simplified as follows:

$$\overline{N}_i = N_i + \Delta P_i = L_i - P_i \tag{9}$$

where $\overline{N}_i$ is a lumped ambiguity in the forms of steps when clock jumps exist. For example, in Figure 3 there are 31 epochs needing to be handled for compensating ambiguity, while for the traditional method nearly 700 epochs need to be handled.

### 2.2. Solution for Type I Clock Jump

This type of clock jump is very common with the observation data download from current IGS data center. Although this type of clock jump does not affect the cycle slip detection, reasonable process noise should be provided in order not to be troublesome. In Kalman filtering, the process noise plays the role to balance the accuracy of state transition for each parameter, so corresponding process noise matrix setting mainly depends on the accuracy of state transition for all parameters, and the ideal situation is that the state transition parameter is exactly correct, for example the ambiguity keeps constant and its state transition parameter is 1 with process noise of zero, so is the position parameter in static mode. There is an example for state transition and process noise matrixes which corresponds to Equation (3), as shown in Equations (10) and (11) when static mode is selected and time gap is 30 s. Both of them are diagonal matrixes, and it can be known from Equation (10) that the state transition matrix is an identify matrix here, since the $t_r$, $ZTD$, and $SION$ are not constant, their corresponding noises are not zero in Equation (11), and it can be seen that the noise of $t_r$ is much larger than other parameters; here, the $9.0 \times 10^{10}$ m$^2$ means the clock may have a $3.0 \times 10^5$ m change between adjacent epochs, and this value is just about 1 ms clock jump. If a small value is given for $t_r$, it means a tight constraint for its variety, and the updated clock error $\tilde{t}_r$ may cannot absorb all the clock jump completely, which leads the coordinates, tropospheric, and ionospheric parameters may absorb part of the residual jump too; for example, the ionospheric delay is not as a constant parameter with both static and kinematic mode, and also the position parameter with kinematic mode in data processing. Assuming that the large process noise is provided for $t_r$, but the process noise of other parameters is usually small, it is easy to obtain a singularity for $Q_{k+1}$, and this disadvantage is inherited by $P_{k+1}^-$ according to Equation (1); as a result, after update, the diagonal elements of $\tilde{P}_{k+1}$ may appear as negative values for some program platforms.

$$\Phi_{k+1} = \begin{bmatrix} 1 \\ & 1 \\ & & 1 \\ & & & 1 \\ & & & & 1 \\ & & & & & 1 \\ & & & & & & \ddots \\ & & & & & & & 1 \\ & & & & & & & & 1 \\ & & & & & & & & & \ddots \\ & & & & & & & & & & 1 \\ & & & & & & & & & & & 1 \\ & & & & & & & & & & & & \ddots \\ & & & & & & & & & & & & & 1 \end{bmatrix} \tag{10}$$

$$Q_{k+1} = \begin{bmatrix} 0 \\ & 0 \\ & & 0 \\ & & & 9 \cdot 10^{10} \\ & & & & 3.3 \cdot 10^{-6} \\ & & & & & 0.01 \\ & & & & & & \ddots \\ & & & & & & & 0.01 \\ & & & & & & & & 0 \\ & & & & & & & & & \ddots \\ & & & & & & & & & & 0 \\ & & & & & & & & & & & 0 \\ & & & & & & & & & & & & \ddots \\ & & & & & & & & & & & & & 0 \end{bmatrix} m^2 \tag{11}$$

In response to the above problem, it is suggested that the prior (predicted) value $t_r^-$ is computed from every epoch's single point positioning (SPP) of pseudo range. If the prior $t_r^-$ is not propagated from the previous epoch, its prior variance cannot be propagated from the previous epoch either. The priori covariance matrix $P_{k+1}^-$ of all parameters to be estimated in PPP can be expressed as Equation (12), where $p_A$ is a $3 \times 3$ matrix of position covariance propagated from the last epoch's a posteriori covariance $\widetilde{P}_k$ while in a static PPP processing without the need to add process noise. $p_D$ is a covariance matrix of the troposphere, ionosphere (for undifferentiated and uncombined PPP ), and ambiguity propagated from last epoch's posterior (updated) covariance, for troposphere and ionosphere small process noise should be considered depending on the time gap of the adjacent epochs, for the ambiguity the processing noise is zero, and $p_B$ and $p_C$ are also composed by the corresponding elements copied from last epoch's posterior covariance, $0_A$ is a $3 \times 1$ matrix full of zeros, $0_B$ is a $1 \times 3$ matrix full of zeros, $0_C$ and $0_D$ are row and column vectors full of zeros, respectively.

$$P_{k+1}^- = \begin{bmatrix} p_A & 0_A & p_B \\ 0_B & p_{t_r}^- & 0_C \\ p_C & 0_D & p_D \end{bmatrix} \tag{12}$$

Now in the $P_{k+1}^-$, the only element that needs to be determined is $p_{t_r}^-$, which denotes the prior variance of $t_r^-$. To obtain the reasonable $p_{t_r}^-$, the simplest way is the SPP via pseudo range, the model can be IF or uncombined which depends on the model of PPP. Of

course, fault detection and exclusion should be implemented for this positioning, and also, in order to reduce the iteration, initial values of SPP can be obtained from $\widetilde{X}_k$. Generally, $p_{t_r}^-$ obtained from SPP is an approximate value due to the low accuracy of the pseudo range. For most situations, this variance can reach a few meters' accuracy; in order not to be overly optimistic, a square of tens of meters is usually a reliable choice, though other better values may exist. aIt should be noted that when the geometric dilution of precision (GDOP) is large, this variance should be enlarged too. For this method, the $p_{t_r}^-$ is directly given, so there is no need to consider how to precisely determine the process noise, and at the same time, the numerical problem can be avoided.

*2.3. Solution for Type II Clock Jump*

Unlike the previous type of clock jump, this one affects the cycle slip detection. Since in the PPP, *MW*, and *GF* combinations are always used as cycle slip detectors. However, from the above discussions, it can be known that the *GF* is unaffected under clock jump. Therefore, additional attention should be paid to the *MW*. In order to avoid the misjudging of cycle slips based on the *MW* under the clock jump, firstly the *MW* filter algorithm is reviewed as Equations (13) and (14) [15]:

$$\langle MW \rangle_{k+1} = K\langle MW \rangle_k / (K+1) + MW_{k+1}/(K+1) \tag{13}$$

$$\sigma_{k+1}^2 = K\sigma_k^2/(K+1) + (MW_{k+1} - \langle MW \rangle_k)^2/(K+1) \tag{14}$$

where $\langle \cdots \rangle$ denotes the mean of *MW*, $k+1$ and $k$ denote current epoch and last epoch, $\sigma$ is the corresponding standard deviation of $\langle MW \rangle$; here, the $K$ denotes order in the satellite data arc.

Before handling the *MW*, whether the type II clock jump happened needs to be judged. The condition to satisfy the Formula (15) can be considered as a clock jump:

$$\begin{cases} Abs(L(k+1) - L(k) - (P(k+1) - P(k))) > 0.001 * c - 15 \\ \\ Abs(GF(k+1) - GF(k)) < \lambda_{ion}/2 \end{cases} \tag{15}$$

where 0.001 denotes the 1 ms and c is light speed, the 15 m is subtracted for the pseudo range noise, $\lambda_{ion}$ means a relaxed constraint for cycle slip exclusion which is the ionospheric wavelength, and for GPS $\lambda_{ion}$ equals 0.054 m when first and second frequencies are used. $Abs(\cdots)$ means the absolute value. The clock jump size *msjp* can be determined by the following formula:

$$msjp = Round(\langle L(k+1) - L(k) - (P(k+1) - P(k)) \rangle / c / 0.001) \tag{16}$$

where the $\langle \cdots \rangle$ denotes the average value for corresponding satellites.

Once the clock jump happens and the size is determined, the *MW* of current epoch can be changed as follows and then Equations (13) and (14) can be used for cycle slip detection.

$$MW(k+1) = MW(k+1) - msjp * c * 0.001 \tag{17}$$

According to Equation (9) and Table 1, the priori ambiguity should be updated by the correction of clock jump if no cycle slip is happening or cycle slip repairing is in need.

$$N_i^-(k+1) = \widetilde{N}_i(k) + msjp * c * 0.001 \tag{18}$$

where the super script $^-$ means the prior value during the filter, $\widetilde{N}_i(k)$ here denotes the corresponding posterior ambiguity from the last epoch, but when cycle slips happen with no repairing and satellite new appearing, there is no need to proceed the correction for prior ambiguity, because this is proceeded by initializing the ambiguity.

After cycle slip detection, the *MW* should be corrected back to the original value in order not to affect next epoch's cycle slip detection, and of course, the $\langle MW \rangle$ should be corrected by the adding of *msjp* ms' distance.

As the above steps are for uncombined models, the IF model is generally the same except for the ambiguity, but it is easy to extend the Equation (18) as follows:

$$N_{IF}^-(k+1) = \widetilde{N}_{IF}(k) + msjp * c * 0.001 \tag{19}$$

Compared to the original observations repairing method, the ambiguity compensating method has an equivalent result, but this new method has the advantage that only the clock jump epoch needs special handling here.

## 3. Results

### 3.1. Receiver Clock Jump Handling Procedure

For receiver clock jump handling in PPP, the procedure is as shown in Figure 4. There are four parts in Figure 4. In the first part, the main task is to judge the type of clock jump and make some preprocessing, and these works are before the Kalman filter. In the second part, the predicted value and variance are provided by a new method. There is no extra handling in the third part. Post-processing is in the last part. In the flowchart, the *msjp0* denotes the corresponding distance of *msjp* ms.

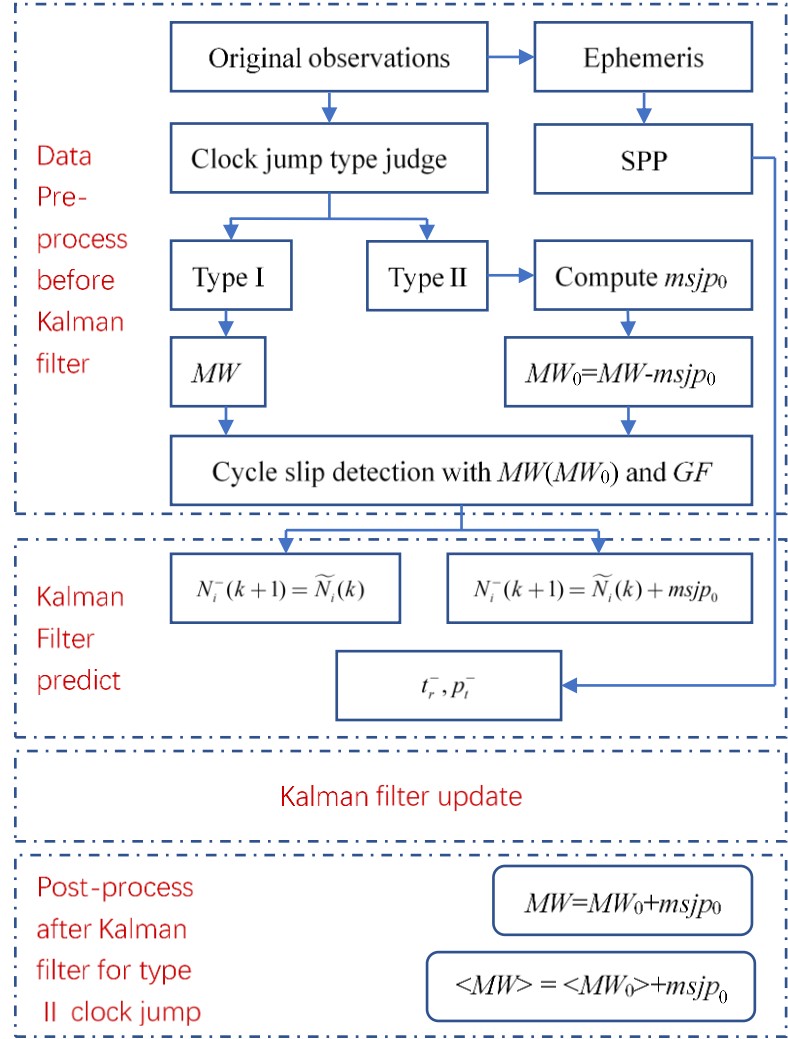

**Figure 4.** Clock jump handling in PPP.

The above procedure is implemented in a high-precision positioning and orbit determination software which considers all the error sources described in the publication [16]. The software adopts undifferentiated and uncombined PPP, both of the static and kinematic GNSS data can be processed by it, and only a forward Kalman filter which is suitable for real-time PPP is used in following validation examples to check the convergence time of the parameters. The default $p_{t_r}^-$ is 625 m$^2$ for GPS, Galileo and BeiDou Navigation Satellite System (BDS), and the corresponding value for Russian Globalnaja Nawigazionnaja Sputnikowaja Sistema (GLONASS) is 2500 m$^2$ for its poor number of satellites; if the GDOP is larger than 6, this variance is enlarged automatically according to the GDOP.

In order to check the feasibility and effectiveness of the new method, for clock jumps of type I, there are two methods to deal with $t_r$: the first one, is the above proposed method, and it can be called A; and the second one is that the predicted value and variance of $t_r$ are propagated from the last epoch, and also the process noise of 10,000 m$^2$ is added as a test, which can be called B. For Method A and B, both static and kinematic modes are validated. For clock jumps of type II, there are also two occasions to deal with the clock jump: the first time is again the proposed method; and the second one is to let nothing affect the clock jump, However, the priori value and variance of $t_r$ are computed by the proposed method, and also both kinematic and static modes are tested.

### 3.2. Validation for the Type I Clock Jump Solution

In this example, GNSS observations on 4 July 2022 at the site ULAB were selected for the case study. The site ULAB is established with the receiver of JAVAD TRE_3 which can receive the signals from GPS, Galileo, GLONASS, and BDS, and the sampling rate is 30 s. After the examination of the raw observations for the four constellations it can be found that all of the carrier phase and pseudo range suffered from receiver clock jumps at the same scale three times (GPS time: 03:08:00; 12:21:00; 20:55:30), and also there is a data gap for every constellation due to unknown reasons. Since the Information and Analysis Center of Navigation (IAC) provides multi-GNSS products, the IAC product is used as the PPP datum. All the constellations have the same receiver clock, and here the Galileo is chosen as an example. From the result, it is clear to see the clock jumps in the time series, as shown in Figure 5.

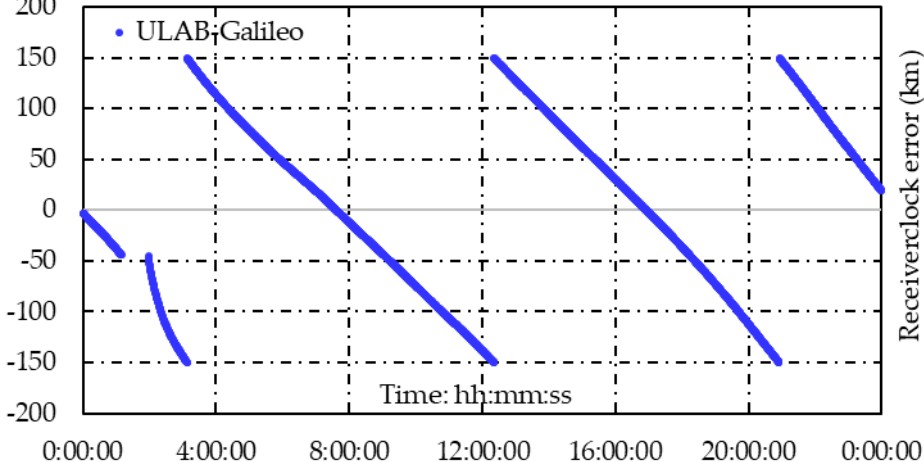

**Figure 5.** Receiver clock error from PPP with Galileo.

An arc of *MW* with E01 is in shown in Figure 6. In Figure 6, during the arc period, there is only one clock jump event at time 03:08:00, and the *MW* has no jump at this time which is the key character of the clock jump of type I.

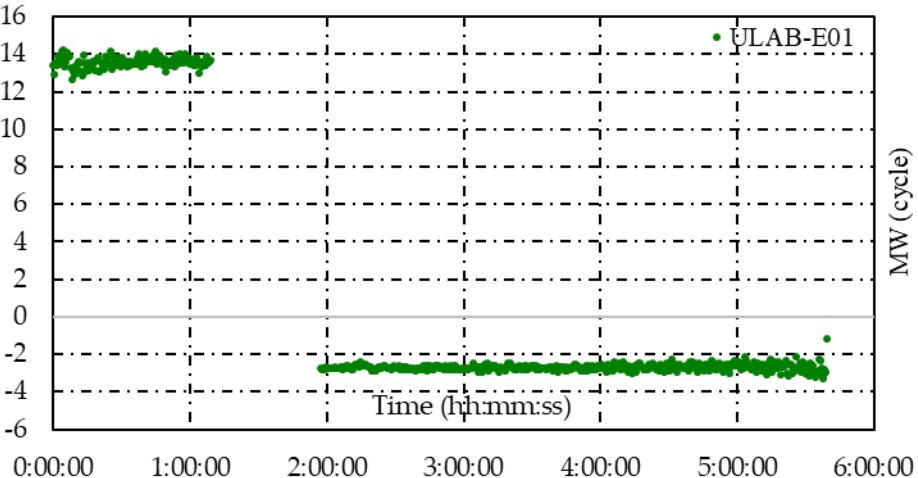

**Figure 6.** Original *MW* arc of E01 for site ULAB.

Usually, for most users, the accuracy of coordinates is the focus. Figure 7 shows the coordinates biases with IGS final coordinates from method B in static mode, and it implies that clock jump of type I hardly affects the position estimation under the default process noise setting. Indeed, coordinates from method A and B are generally the same.

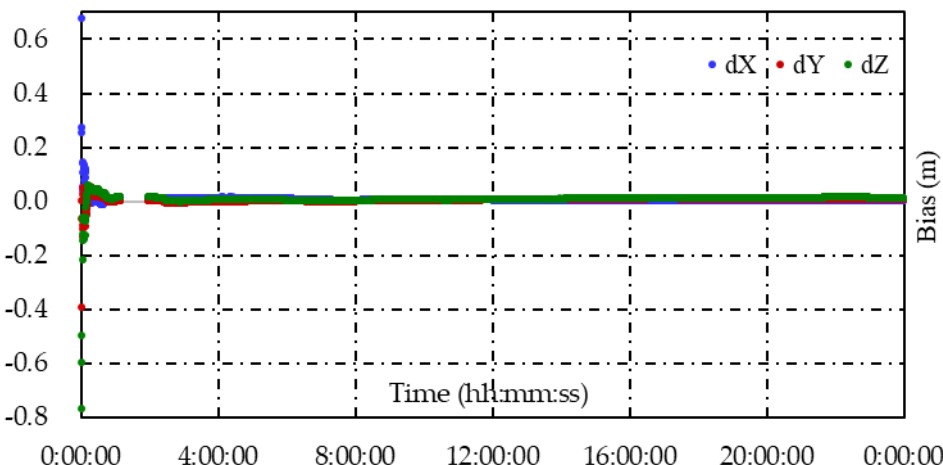

**Figure 7.** Coordinate biases from static PPP with method B.

The ZTDs from static PPP with method A and B are shown in Figure 8. From Figure 8 it can be seen that the ZTDs time series are smooth for both methods except during the gap time. The convergence time of each method is in half an hour, which is within a normal area. After convergence, the ZTD results are highly consistent, and there is no sudden jump at the clock jump times with method B, which implies that the clock jump of type I hardly affects the ZTD estimation under the current process noise setting.

For the uncombined PPP, the ionospheric delay is a by-product, though this original slant delay contains hardware delay, it can be used for further application such as ionospheric tomography. The difference of slant ionospheric delay (dSION) between method B and A from time 20:50:00 to 21:00:00 is analyzed; during this time period there are eight Galileo satellites observed, and at time 20:55:30, a clock jump event happens, the position, ambiguities and tropospheric delay have a good convergence at this period, the dSIONs are shown in Figure 9. From Figure 9 it can be seen that there is a same-scale sudden jump which is about 4.5 mm at time 20:55:30 for each satellite, and this means even in static PPP, if no reasonable process noise is provided for receiver clock error, the ionospheric delay result will be degraded too. It seems that only one epoch is affected here, but if smaller

process noise is set, more epochs are affected. Additionally, there is also a shift of about 2 cm for each ionospheric delay series.

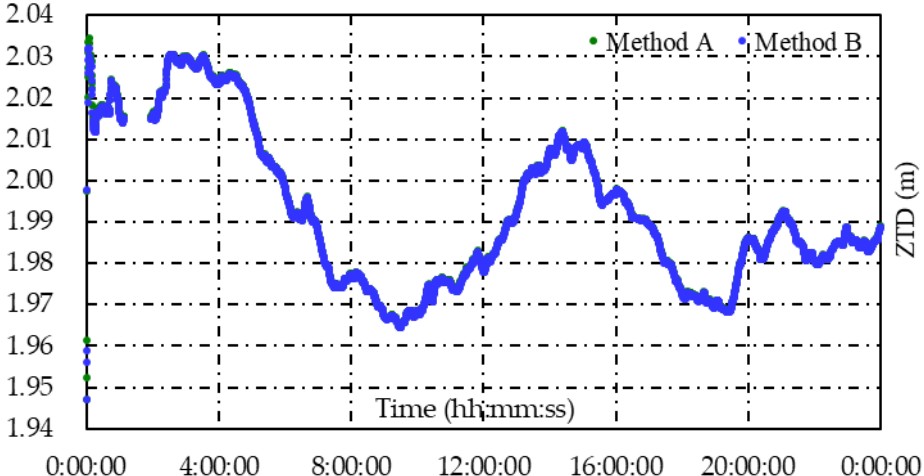

**Figure 8.** ZTDs from static PPP with methods A and B.

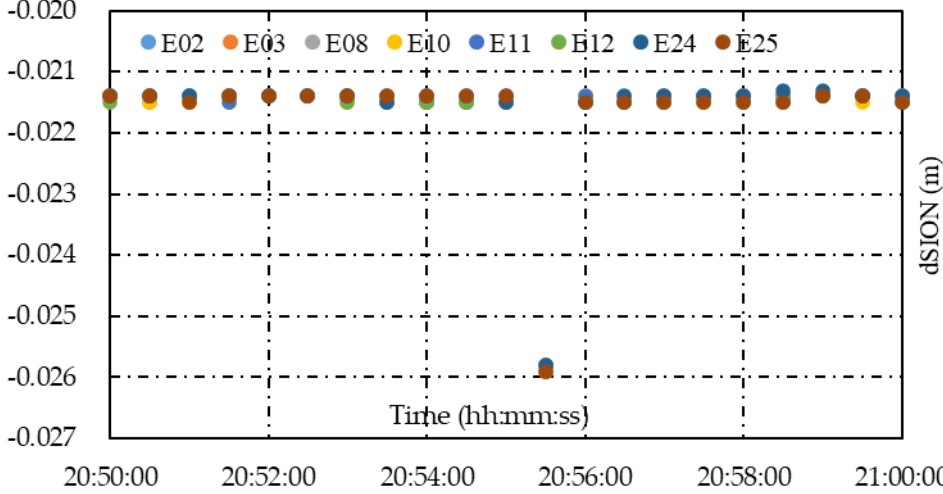

**Figure 9.** dSIONs between method B and A from static PPP with various satellites.

In static PPP, the process noise of coordinates is zero, but in kinematic mode, their process noise is usually larger than that of tropospheric delay and ionospheric delay, so position parameter may be affected by clock jumps of type I. Coordinates biases between method A or B and IGS final coordinates are as shown in Figures 10 and 11. Though the results in Figures 10 and 11 are similar, the accuracy of both methods is within 1 dm after convergence, but there are differences at times 03:08:00, 12:21:00, and 20:55:30, and there are a few cm jumps, so a part of clock jumps of type I are compensated by coordinates in kinematic mode for method B.

The ZTDs in kinematic mode has a wide application prospect [17]. Compared to position, ZTD has a smaller process noise setting which is 2 cm/sqrt(hour), the same as static mode. As seen in Figure 12, the ZTDs from method A and B are generally the same, and both of them have longer convergence time and larger fluctuation compared to the static mode; again, the clock jump of type I has little influence on ZTD estimation in kinematic mode.

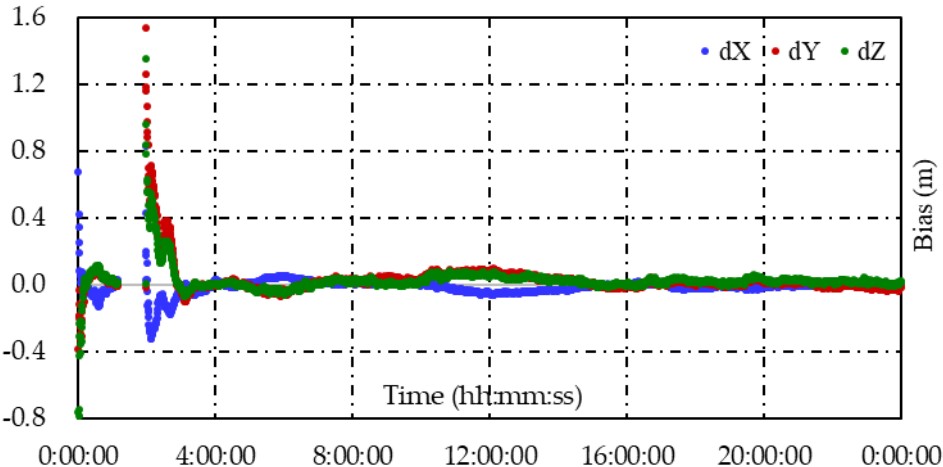

**Figure 10.** Coordinate biases from kinematic PPP with method A.

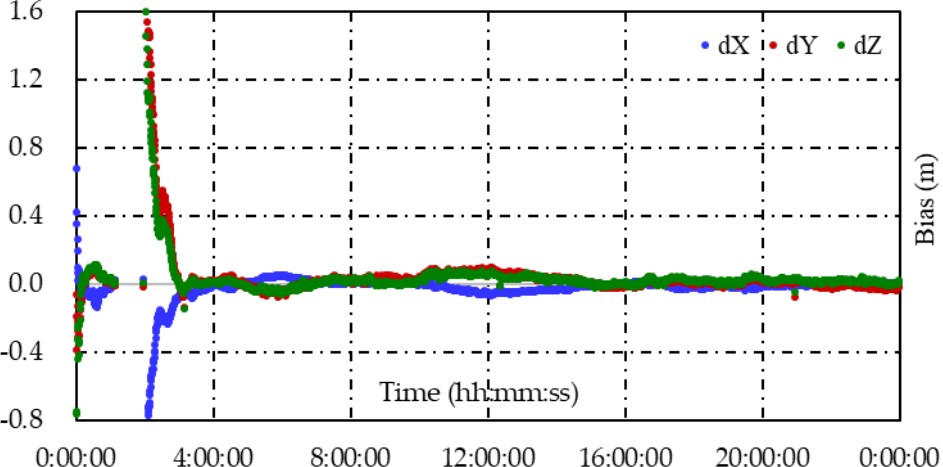

**Figure 11.** Coordinate biases from kinematic PPP with method B.

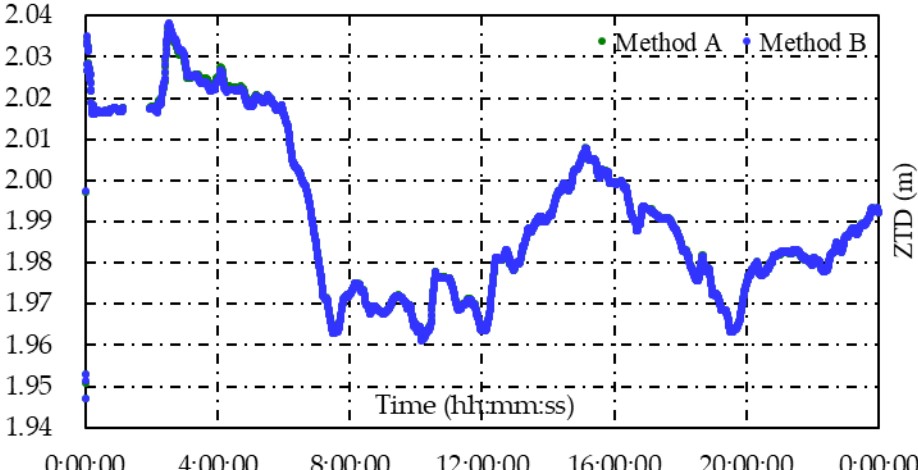

**Figure 12.** ZTDs from kinematic PPP with methods A and B.

Similar to the static PPP result, the dIONs between method B and A from kinematic PPP with various satellites are shown in Figure 13. As seen in Figure 13, there are sudden jumps for all the ionospheric delay series at time 20:55:30, but unlike the static PPP result, the jumps are not in a same scale here, ranging from nearly 2 cm to less than −2 cm. Additionally, there is a shift of about −4 mm for the ionospheric delay of each satellite.

Compared to static mode, in kinematic mode the ionospheric delay may absorb less clock jump due to larger prior variance of position.

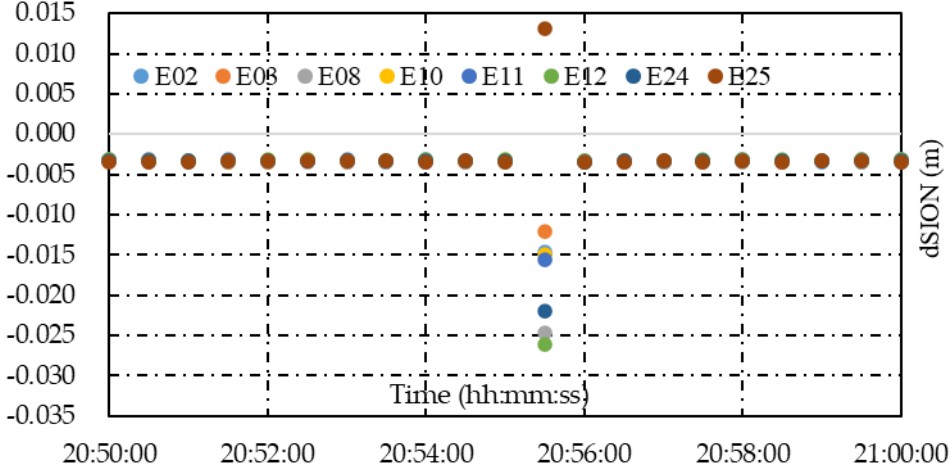

**Figure 13.** dSION between method B and A from kinematic PPP with various satellites.

### 3.3. Validation for Type II Clock Jump Solution

For this case, the aforementioned site PIXI is selected. On 12 May 2008, the site suffered an earthquake of magnitude 8.0, and it was about 48 kilometers far from the epicenter. In the case of earthquake alarm application, PIXI receives the GPS signal at 1s intervals. After examining the PIXI observation, it can be found that the pseudo ranges suffered from ms jump (MSJP) every 674 s, but the carrier phase is smooth, and the observation data are interrupted by a power outage due to the big earthquake, and the last time to record observation is at 6:29:10 (GPS time, the same as the following), and the earthquake happened at 6:28:17, so there was about 1 min to capture the kinematic deformation despite the earthquake lasting 80~120 s. At this time, the IGS final orbit and clock are used as the PPP datum. From Figure 14, it is clear to see the MSJP in the time series. An *MW* arc of G02 is shown in Figure 15; unlike in Figure 6, this *MW* has a jump every 674 s. If unaltered, corresponding ambiguities under this clock jumps of type II will be reinitialized about every 11 min, and of course the accuracy of all parameters will be seriously degraded since it cannot benefit from the accurate ambiguities in this situation.

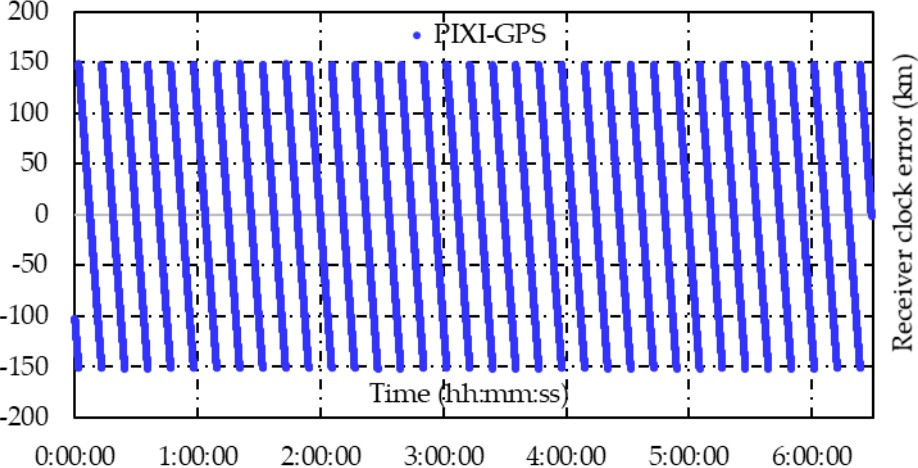

**Figure 14.** Receiver clock error of PIXI with GPS.

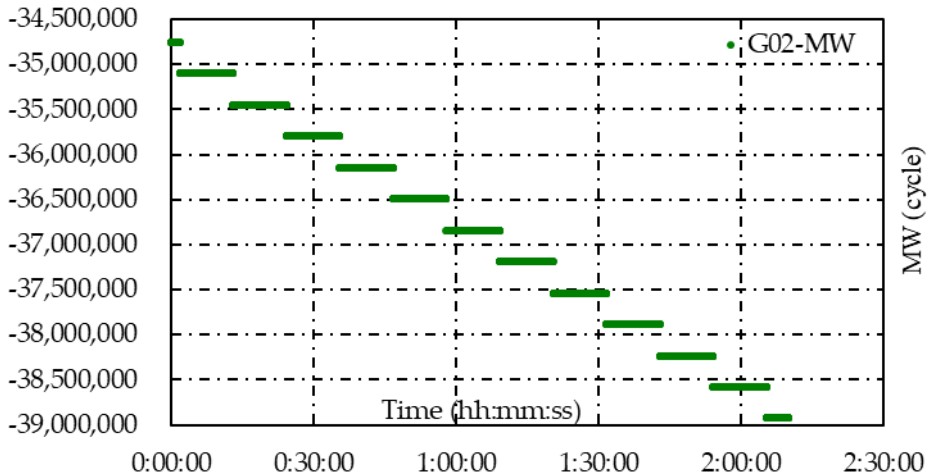

**Figure 15.** Original *MW* arc of G02 for site PIXI on 12 May 2008.

Firstly, the kinematic mode is selected. Compared to static solution coordinates of Canadian Spatial Reference System Precise Point Positioning (CSRS-PPP) with the first 6 h observation, coordinates biases from kinematic PPP without and with MSJP correction are as shown in Figures 16 and 17, respectively; during the 350s from time 6:23:20 to 6:29:10 there is only one MSJP event, which happens at time 6:23:35. It is clear to see in Figure 16 there is an initialization at time 6:23:35, the dY has the biggest jump since it mainly represents the up component, and also there is a reconvergence process from time 6:25:30 to 6:28:00. The biases in Figure 17 are much smoother compared to Figure 16, which means that after the MSJP correction there is not any impact on position estimation, and the biases have an obvious deformation at time 6:28:31; this is reasonable because the earthquake needs time to arrive PIXI, and it can be seen that the max deformation reaches 1 m, which is consistent with the result of double-difference-based positioning [18]. Though biases during the last 40 s in Figure 16 are similar to that in Figure 17, the largest difference exceeds 1 dm, which goes against for earthquake analysis.

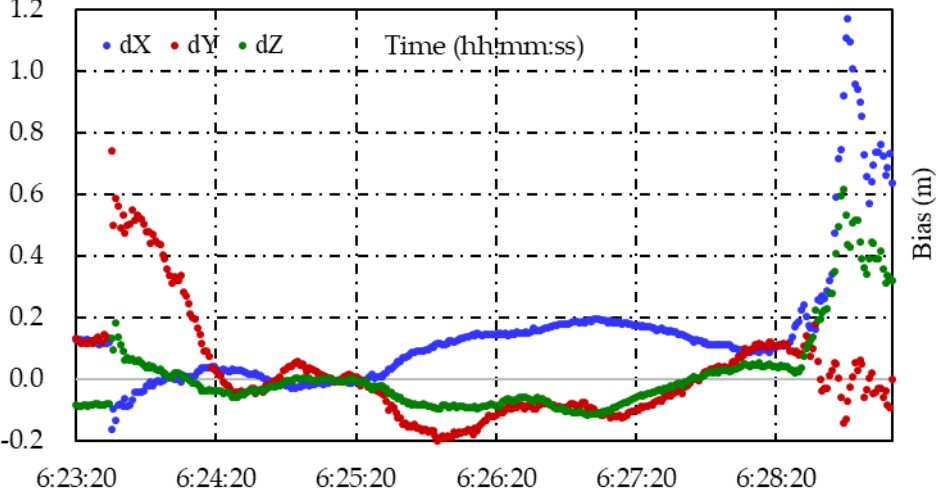

**Figure 16.** Coordinate biases from kinematic PPP without MSJP correction.

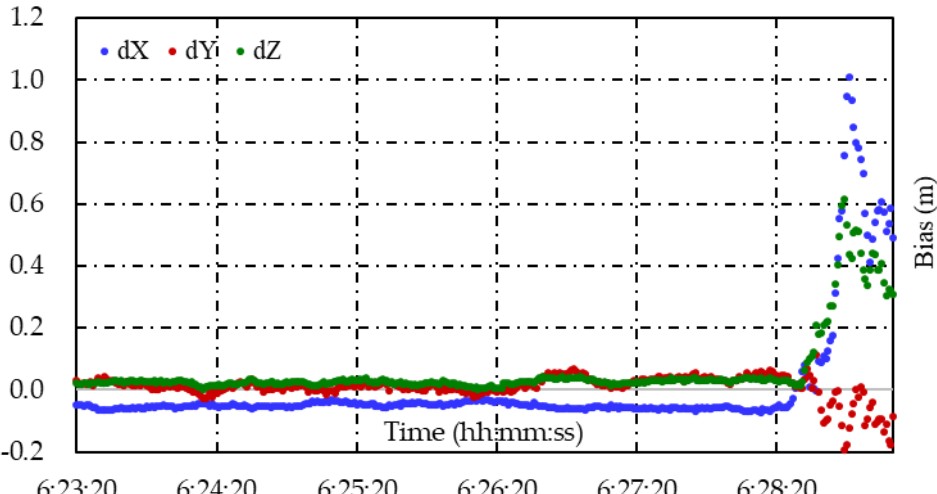

**Figure 17.** Coordinate biases from kinematic PPP with MSJP correction.

The ZTDs of PIXI from kinematic PPP with and without MSJP correction are shown in Figure 18. Unlike the clock jump of type I, which hardly affects the ZTD estimation, the difference in ZTDs between with and without MSJP corrections in kinematic PPP is quite big, and it even exceeds 8 cm, though there is not any sudden jump in the blue ZTD line, which means that similar to the position, clock jumps of type II have a great impact on ZTD estimation too.

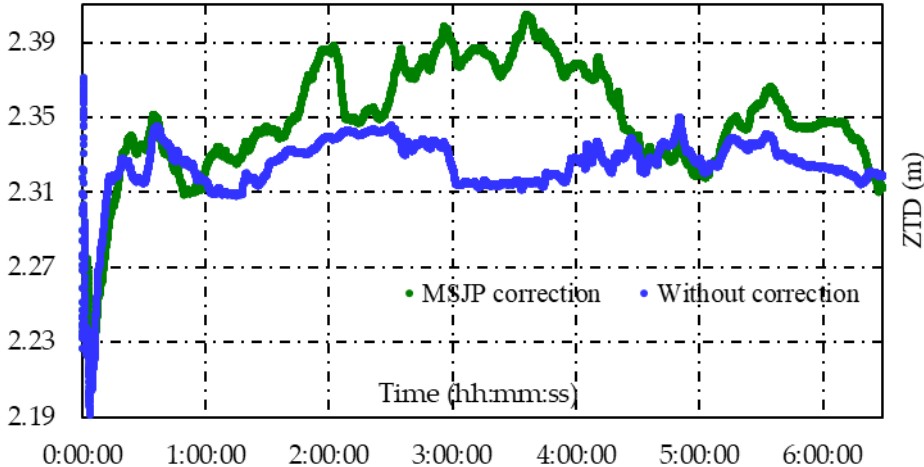

**Figure 18.** ZTDs from kinematic PPP with and without MSJP corrections.

The corresponding dSIONs are shown in Figure 19. From Figure 19 it can be seen the largest dSION, which exceeds −0.14 m, is shown on satellite G09, and the max dSION of another satellite G14 is close to 0.13 m, and the dSION of other satellites are within 1 dm. Additionally, there are turns at time 6:23:35 on G21 and G30, though they are not obvious, but they are affected by the accuracy of the new ambiguities indeed. The above difference implies that the clock jumps of type II have a great impact on ionospheric delay estimation in kinematic PPP.

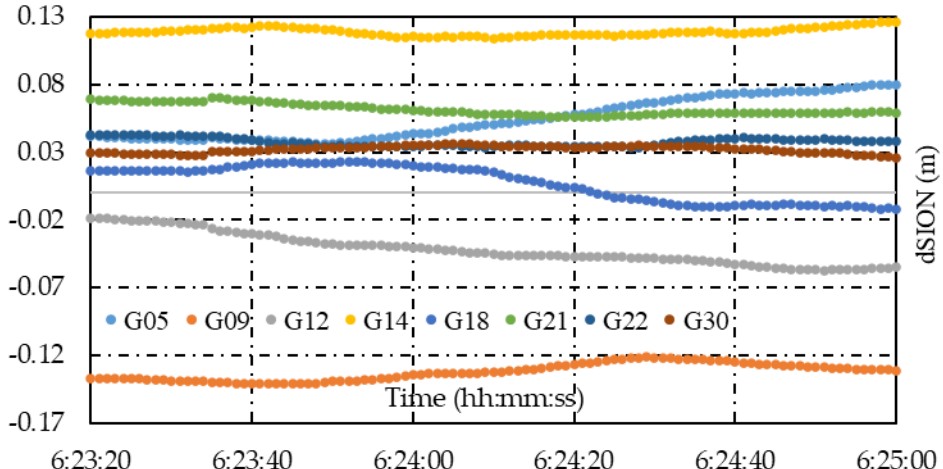

**Figure 19.** dSIONs between kinematic PPP without and with MSJP correction.

It is obvious to see the improvements by the proposed method from the above results in the kinematic mode. In order to analyze the improvements in static mode, the corresponding first 6 h' observation of PIXI is adopted assuming that there is no slip before earthquake. Again, the final coordinates of CSRS-PPP solution are as reference, and the coordinates biases without and with MSJP corrections are shown in Figures 20 and 21. The final biases in Figure 21 are at the mm level in all components, compared to that 7 cm in dY in Figure 20, and also the convergence is much worse in Figure 20, which takes nearly 3 h to reach 1 dm accuracy, while the first time to reach this accuracy in Figure 21 is 21 min after beginning. It should be noticed that the influence level of this clock jump mainly depends on the frequency of the clock jump event.

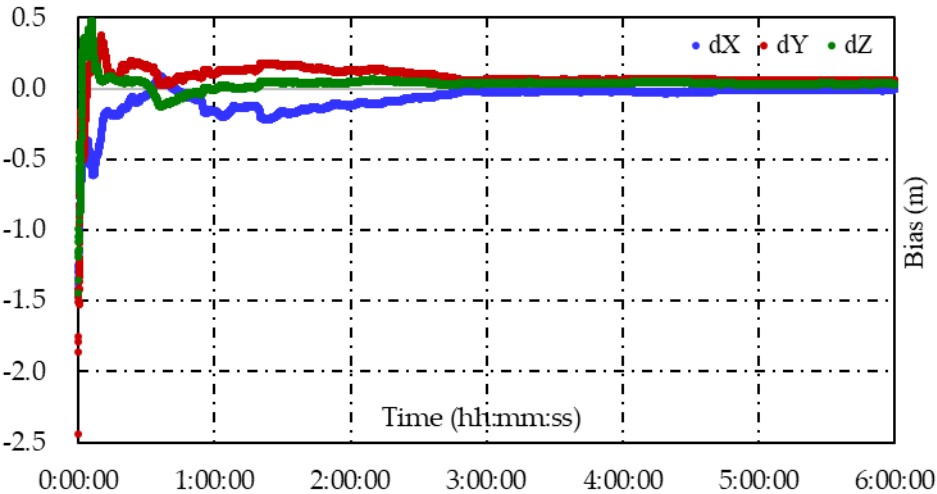

**Figure 20.** Coordinate biases from static PPP without MSJP correction.

From previous analysis, it can be known that the clock jumps of type II have great impact on position estimation in static mode. Compared to the state transitions of coordinates, which are fixed in static mode, the state transitions of tropospheric delay and ionospheric delay are more random, especially they are highly related with the height component; if the height cannot be solved well, they may be obtained with bad results too. The ZTDs from static PPP with and without MSJP corrections are shown in Figure 22, and according to Figure 22, the difference between the two series can exceed 6 cm, which is a little smaller than that in the kinematic mode, but this difference cannot be ignored for meteorological applications.

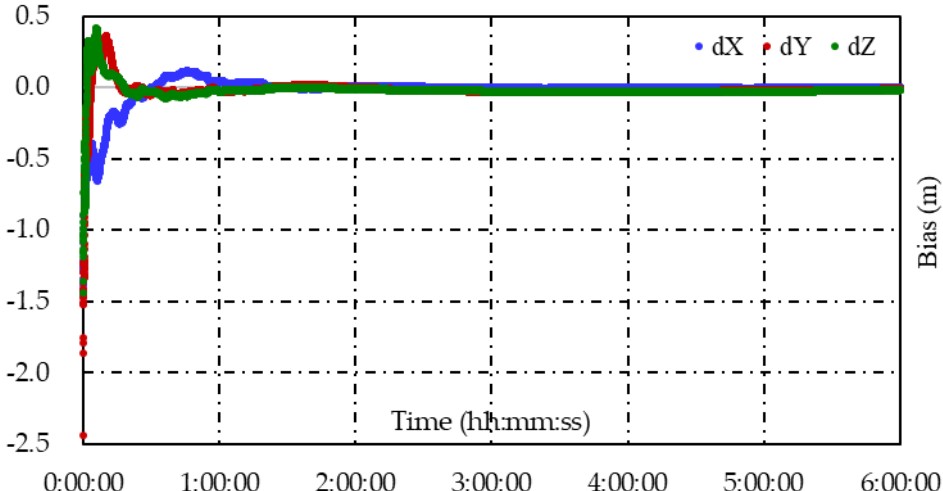

**Figure 21.** Coordinate biases from static PPP with MSJP correction.

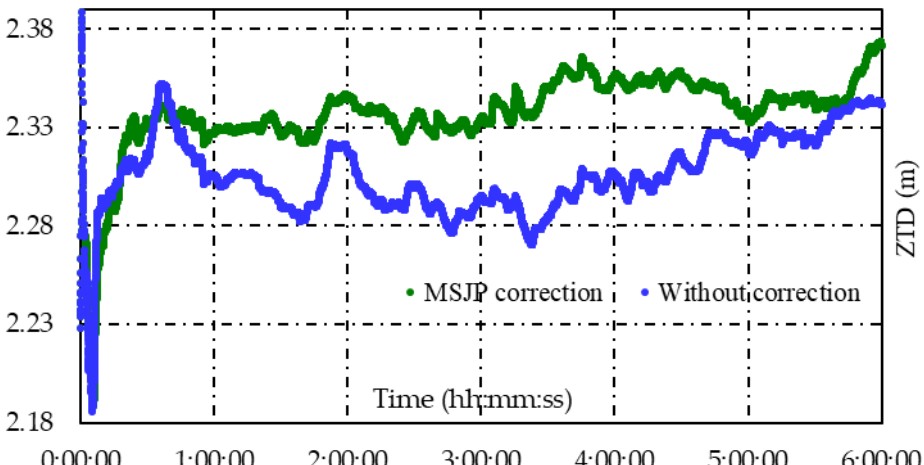

**Figure 22.** ZTDs from static PPP with and without MSJP corrections.

The dSION of each satellite from time 5:49:10 to 5:50:50 in static PPP is shown in Figure 23; during this period, there is one clock jump event which happened at time 5:49:59. Compared to the kinematic mode, the scale of dSION is smaller in static mode, which is about two-thirds of the former. There are turns at time 5:49:59 on satellites G05 and G22 too.

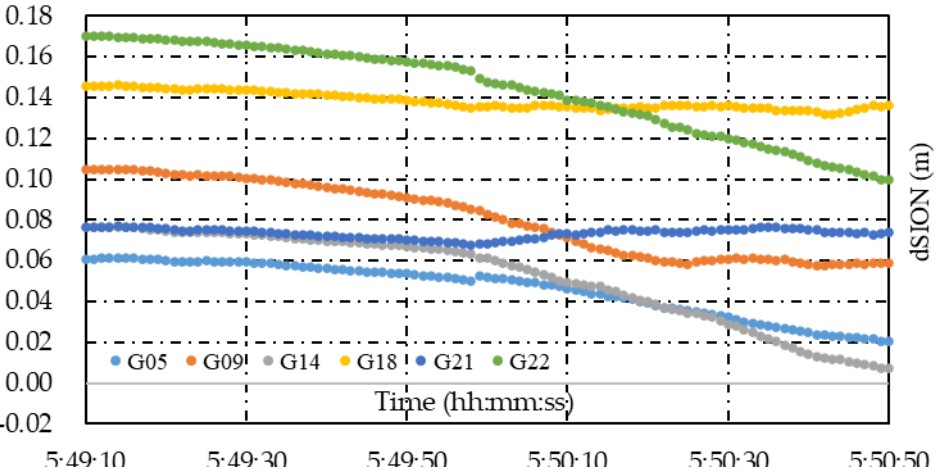

**Figure 23.** dSIONs between static PPP without and with MSJP corrections.

## 4. Discussion

The clock jump of type I is usually ignored in most cases, since it does not affect cycle slip detection and repair. In some robust software, the process noise of $t_r$ is usually very large, if a special mathematic algorithm is not implemented it will cause the covariance matrix singularity since the process noises of the other parameters is very small, and the program platform may have a limited precision. Here, a process noise of 10,000 m$^2$ is given for $t_r$, and under this setting, the ionospheric delay is affected with a few mm or cm in kinematic or static mode; also, sudden jumps exist at the clock jump epoch and so do the position parameter in the kinematic mode. If smaller process noise is provided, the result is much worse, as seen in Figures 24 and 25. If a larger one is provided, such as 90,000 m$^2$, the numerical problem arises, so the new method is used here.

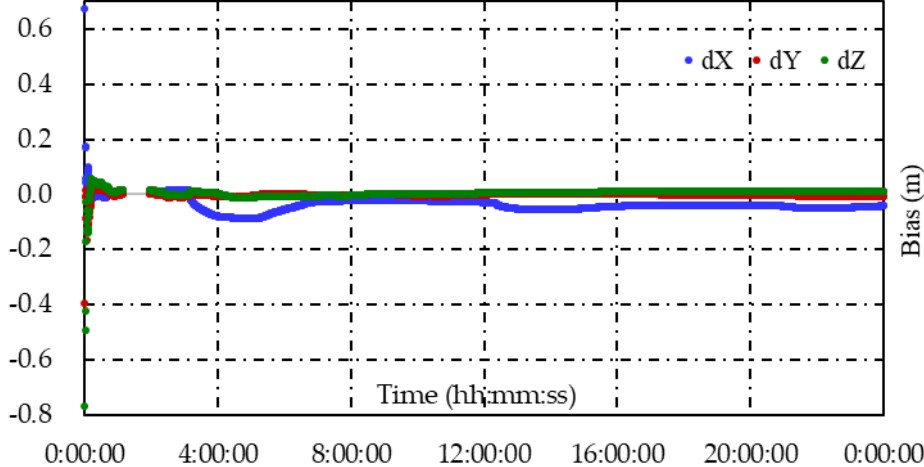

**Figure 24.** Coordinate biases from static PPP with method B with respect to Figure 7 (process noise of 100 m$^2$ is set for receiver clock error).

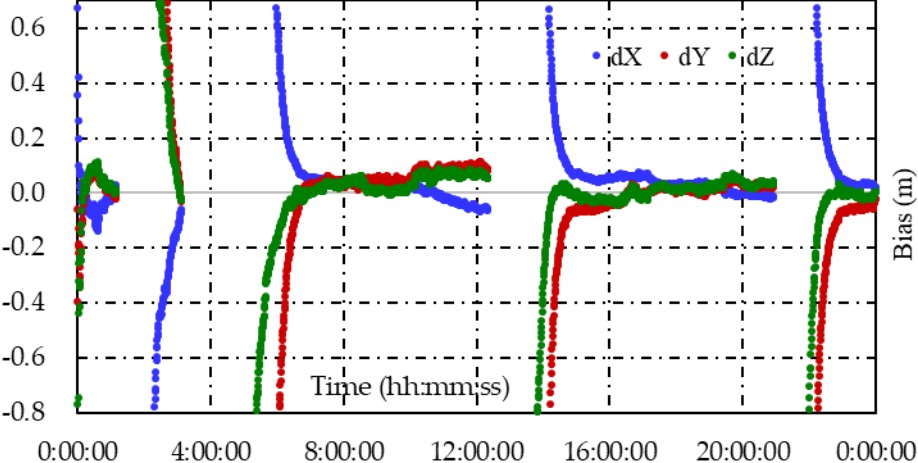

**Figure 25.** Coordinate biases from kinematic PPP with method B with respect to Figure 11 (process noise of 2500 m$^2$ is set for receiver clock error).

For the clock jump of type II, take the PIXI as an example; if the traditional method is adopted, all the carrier phase should be correct after epoch 110. Since the total number of epochs is 23349, both L1 the L2 need to be corrected, so the total number of the correction is $2 \times (23{,}349 - 110) \times 7 = 325{,}486$ if an average of seven satellites are observed. While the number of correction for new method is related to the number of clock jump events, which is 35 in the test, the total number of corrections is $(7 + 14 + 7 + 7) \times 35 = 1225$, where the first 7 means number of $MW$, the 14 means number of ambiguities (uncombined PPP, for

IF combination is 7), the second 7 is the number of *MW* that need to be corrected back, and the last 7 denotes the number of $\langle MW \rangle$. Additionally, the traditional clock jump correction exists in the form of steps, so previous clock jumps should be saved for the adding up work which also consumes extra computation burden.

For different receiver clocks, the frequency of clock jump events is quite different, and some sites have a few times one day, while others may have more than 70 one day, such as the IGS sites POTS and WIND during early 2022. The number of events directly affects the total quality of results because the ambiguity needs time to convergence, and a few minutes are of course not enough.

## 5. Conclusions

In this article, the quantities related to receiver clock jumps are presented, and the relationship among them is analyzed. According to the characteristics of the receiver clock jumps, they can be divided into two types. The first type is that the pseudo range and carrier phase have the same scale jump and the second one is that they have different scale jumps. Whatever the type of receiver clock jumps, it is proposed to use the SPP of pseudo range to obtain a priori variance of the receiver clock error for every epoch, and since the process noise of receiver clock error is unpredictable, the corresponding variance can be given with a relatively large value in order to avoid the singularity of the process noise matrix. If the type II clock jump happens, it is proposed to handle the *MW* and ambiguity instead of handling so many original observations. The new method is more effective and yet easier to implement than the existing methods since all the of raw observations at every epoch should be corrected after the clock jumps to compensate the receiver clock jumps.

After the numerical validation of the new method, it shows that the clock jump of type I has a cm-level impact on ionospheric delay estimation with kinematic and static PPPs under the default setting of process noise for receiver clock, so does the position estimation in kinematic mode, but if smaller process noise is provided, the results are worse. For clock jumps of type II, impact on tropospheric and ionospheric delay can reach the cm or dm level respectively, so do the coordinates in static and kinematic mode. Based on the validation, the new method has the potential for use in various GNSS data processing, especially in real-time PPP and related applications.

**Author Contributions:** Conceptualization, S.X.; methodology, S.X.; validation, S.X.; formal analysis, S.X.; data curation, S.X.; writing—original draft preparation, S.X.; writing—review and editing, J.L., J.W. and W.Z.; supervision, J.W. All authors have read and agreed to the published version of the manuscript.

**Funding:** This work is supported by the Sichuan Science and Technology Program (NO:2021YFG0339).

**Institutional Review Board Statement:** Not applicable.

**Informed Consent Statement:** Not applicable.

**Acknowledgments:** The authors would like to thank the institutions IGS, Sichuan Earthquake Administration, IAC, and CSRS-PPP for providing the observations, products, and references. Our special thanks go to all the reviewers for their constructive and valuable comments.

**Conflicts of Interest:** The authors declare no conflict of interest.

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
