# Peer review of "An Efficient Method to Compensate Receiver Clock Jumps in Real-Time Precise Point Positioning"

_remotesensing, doi:10.3390/rs14205222_

Round 1
Reviewer 1 Report
The article describes some procedures to deal with clock jumps affecting receivers' clocks.
It deals with two two types of jump: one on phase and pseudoranges; and the other one only on pseudorange.
It then describes a procedure to deal with the first jump types in the Kalman filter estimation and a procedure to deal with the second type of jumps in a cycle slip detection procedure.
However the used filter is not properly described so it is very difficult to interpret the results; the exposition has also many problems making the manuscript difficult to read.
The deal of clock jump of type one is not properly explained but seems to be related to the choice of the apriori covariance matrix to be assigned to the new clock receiver unknown. However, since the equation of the filter is not properly explained it is difficult to understand what is proposed.
The second method deals with avoiding the detection of cycle slips on MW combination in case of a clock jump of type II. Such a method is compared to a direct correction of phase observables.
The first method is comparedagainst a simple fixed choice of 10000 m2 in the process noise showing very little difference and so no particular improvement.
The second method is compared against letting the MW marking a cycle slip which makes the model weaker and thus degrades the results.
Overall I believe the methods are poorly explained and the results are not particularly significant.
I still believe that a severely revised version of the article might be publishable, this revision should contain at least:
- A general improvement of the exposition.
- A proper introduction for the used filter ( the unknowns, equations, the stochastic model)
- Some more interesting results. For instance, trying different levels of process noise in case of jump, or different apriori covariance matrix for the new unknowns. Highlighting also the numerical problems that might arise.
For all these reasons I recommend a major revision.
Specific comments:
Equations are frequently referenced before being introduced. This is not a good practice.
Please explain better equation 4 it is the phase geometry free combination?
Table 1 is unclear, is one column missing?
Line 133,139 : PIXI CUT0 please introduce better these sites; are they IGS stations? Which equipment do they use?
Line 165: n with the observation data download from 165 current IGS data centers since they are smoothed before upload -> It is not clear to me to which smoothing you are referring
Line 170: and ionospheric parameters may absorb part of the residual jump too -> Ionosphere unknown should be independent of clock troposphere and coordinates please elaborate better on why it is influenced
Section 2.2 needs a proper introduction of the filter model to be understandable
It would be good to accompany the results with some statistical index such RMS or quantiles of errors
Reviewer 2 Report
The manuscript focuses on the handing of receiver clock jumps in real-time PPP. This research has important practical value and significance for the performance improvement of real-time PPP.
1. Page 2, Line 66-67. “…, also the old receivers such as the Trimble 5700 may be still in operation due to its good quality”. This sentence is ambiguous and lets readers confused. “good quality” leads to “this problem”?
2. Page 3, Table 1. Please add the row information or unit in Table 1. How to understand and describe Table 1? Below Table 1, what is the correspondence between types 1-4 and table 1?
3. Page 4, Figure 1. The title of Figure 1 is inconsistent with the description in the text as follows “In Figure 1, the original L1 and P1 observations for satellite G02 of site CUT0 have the same scale clock jump”, please check. Besides, time information like “XX-2020-001” or “XX-2008-133”in the Figure is ill-formed and should be removed, please add that information (Year-DoY) in the title of the Figure.
4. Page 5, Line 161-162. “For example, in the Figure 3 there are 31 epochs needs to be handled, while for the traditional method nearly 700 epochs need to be handled.” Please show the results of the traditional method in the form of Figure 3 and more intuitively reflect the advantages of the new method.
5. Page 5, Line 177-178. The sentence can be corrected as “… is computed from every epoch’s Single Point Positioning (SPP) of pseudo range”.
6. Page 6, Line 212-213. “k+1 and k denotes current epoch and last epoch”. “denotes” should be correct as “denote”, and the letter k (lowercase letters) does not exist in Equation (6) and (7), please check. More importantly, there are some grammatical errors in this manuscript, which need to be carefully corrected.
7. Page 7, Figure 4. The flow chart of clock jump handing in PPP is not standardized, and it is better to redraw.
8. Page 8, Line 281-282. “Since Information and Analysis Center of Navigation (IAC) provides multi GNSS product, so the IAC product is used as the PPP datum”. “since” and “so” cannot be used in the same sentence at the same time, please correct similar syntax errors one by one.
9. Page 8, Figure 5. The horizontal ordinate in Figure 5 can be corrected in GPS hours (00:00:00 to 23:59:59). This is convenient to correspond to three specific times of the receiver clock jump. Other Figures in this manuscript need to be similarly modified.
10. Page 17, Conclusions. The results related to ZTD and ionospheric need to be added to the conclusions and at least two paragraphs can be written to summarize the key findings.
